# Interplay between Mg²⁺ and Ca²⁺ at multiple sites of the ryanodine receptor

Ashok R. Nayak [1], Warin Rangubpit [2], Alex H. Will [1], Yifan Hu[1], Pablo Castro-Hartmann[1,5], Joshua J. Lobo [1], Kelly Dryden [3,6], Graham D. Lamb [4], Pornthep Sompornpisut [2] ✉ & Montserrat Samsó [1] ✉

RyR1 is an intracellular Ca²⁺ channel important in excitable cells such as neurons and muscle fibers. Ca²⁺ activates it at low concentrations and inhibits it at high concentrations. Mg²⁺ is the main physiological RyR1 inhibitor, an effect that is overridden upon activation. Despite the significance of Mg²⁺-mediated inhibition, the molecular-level mechanisms remain unclear. In this work we determined two cryo-EM structures of RyR1 with Mg²⁺ up to 2.8 Å resolution, identifying multiple Mg²⁺ binding sites. Mg²⁺ inhibits at the known Ca²⁺ activating site and we propose that the EF hand domain is an inhibitory divalent cation sensor. Both divalent cations bind to ATP within a crevice, contributing to the precise transmission of allosteric changes within the enormous channel protein. Notably, Mg²⁺ inhibits RyR1 by interacting with the gating helices as validated by molecular dynamics. This structural insight enhances our understanding of how Mg²⁺ inhibition is overcome during excitation.

Ryanodine receptors (RyRs), the largest ion channels known (2.2 MDa), mediate intracellular Ca²⁺ release from the sarco/endoplasmic reticulum (SR/ER) in excitable and non-excitable cells. Skeletal muscle, which supports voluntary contraction, expresses RyR1, whereas cardiac muscle expresses RyR2. These two RyR isoforms, together with RyR3, are also expressed in brain and other organs[1–3]. In skeletal muscle, action potentials activate the L-type Ca²⁺ channel (Ca$_V$1.1, also known as dihydropyridine receptor (DHPR)) resulting in RyR1 opening and subsequent muscle contraction, in the process known as excitation-contraction (EC) coupling. The L-type voltage-gated Ca²⁺ channel works as a sensor located in the T-tubule membrane and activates the RyR1 via conformational coupling[4,5]. The RyR pore lacks Ca²⁺ selectivity, allowing co-permeation of other cations such as K⁺, Na⁺, and Mg²⁺ through the open channel[6]. Paradoxically, Mg²⁺, the predominant divalent cation in the cytosol, is a constitutive RyR1 inhibitor. Hence, in the absence of membrane depolarization, muscles are actively kept quiescent by Mg²⁺[7–10]. Mg²⁺ inhibits RyRs by unknown mechanism(s), but when the channel is activated, Mg²⁺ permeates through the RyR1 pore and competes with the Ca²⁺ current[6,11]. Intracellular Mg²⁺ (about 8 mM in total in the cytoplasm) is mainly complexed in the form of ATP-Mg²⁺ (~7 mM), the physiologically active form of ATP, while 0.9–1.5 mM exists as free Mg²⁺[12–14]. Abnormally low free Mg²⁺ cytosolic concentrations result in spontaneous RyR1-mediated Ca²⁺ release[8,9], and impaired regulation of RyR1 by Mg²⁺ confers susceptibility to malignant hyperthermia (MH)[15,16].

Ca²⁺ has a biphasic effect on RyRs[7], which is mediated by a high-affinity, activating Ca²⁺-binding site[17,18] and putative low-affinity Ca²⁺ binding site(s)[19,20]. Thus, an intricate control of RyR gating by its ligands in the physiological scenario emerges, where the concentrations of Mg²⁺ and ATP are kept constant but DHPR conformational change and subsequent increase of cytosolic Ca²⁺ concentration constitute the principal activation variables. How the two divalent cations, Mg²⁺ and Ca²⁺, have such disparate effects on RyR has remained a mystery with several, non-exclusive mechanisms proposed; these include (a) Mg²⁺ competition at the high-affinity Ca²⁺ activation site producing an opposite effect[7,8,21,22], (b) Mg²⁺ and Ca²⁺ binding to low-

¹Department of Physiology and Biophysics, Virginia Commonwealth University, Richmond, VA, USA. ²Department of Chemistry, Faculty of Science, Chulalongkorn University, Bangkok, Thailand. ³Department of Molecular Physiology and Biological Physics, University of Virginia, Charlottesville, VA, USA. ⁴Department of Microbiology, Anatomy, Physiology and Pharmacology, La Trobe University, Melbourne, VIC, Australia. ⁵Present address: ThermoFisher Scientific, Cambridge, UK. ⁶Present address: Department of Chemistry and Biochemistry, UC Santa Barbara, Santa Barbara, CA, USA.
✉e-mail: pornthep.s@chula.ac.th; montserrat.samso@vcuhealth.org

affinity cytosolic inhibitory sites producing a similar effect[7,16,20], and (c) competition for ion flow[6,21,23].

In this work, we determined the cryo-EM structures of RyR1 with its major inhibitory divalent cation $Mg^{2+}$ at two different concentrations which reveal three putative $Mg^{2+}$ binding sites per subunit in the cytoplasmic domain of the channel that are shared with $Ca^{2+}$ and a fourth exclusive binding site situated in the pore pathway. Ion binding of either $Ca^{2+}$ or $Mg^{2+}$ to each of these sites results in conformational changes of RyR1 that depend on the specific type of cation bound at each site. We carried out molecular dynamics (MD) to understand the interaction of $Mg^{2+}$ and $Ca^{2+}$ with the $Mg^{2+}$ binding site at the pore pathway, and the $Mg^{2+}$-induced interaction networks in open and closed states. The transmembrane domain of the RyR1-$Mg^{2+}$ structures is surrounded by intact reconstituted nanodiscs and resolved lipid densities indicative of a membrane-embedded state free from detergent-induced artifacts[24–26]. Our MD-interpreted structures explain the permeation of $Mg^{2+}$ in open channels and inhibition by $Mg^{2+}$ binding to multiple sites. Functional radioligand binding using [3H] ryanodine in the presence of different combinations of $Ca^{2+}$, $Mg^{2+}$, ATP, and AMP-PCP (ACP) illustrate the competing effects of $Mg^{2+}$ and $Ca^{2+}$ on RyR1 and substantiate our findings from structural and MD data. Altogether, this work provides a fundamental insight into the complex regulation of RyR1 by the divalent cations $Ca^{2+}$ and $Mg^{2+}$.

## Results

### $Mg^{2+}$ overrides the activating effects of $Ca^{2+}$ and ATP

We studied the effect of ATP and $Mg^{2+}$ on the $Ca^{2+}$-induced activity profile of rabbit RyR1 using the tritiated ([3H]) ryanodine binding assay, which reflects the probability of channel opening[27]. The assay, carried out under a wide range of $Ca^{2+}$ concentrations, indicated typical

biphasic response of RyR1 to free $Ca^{2+}$ and a three-fold upregulation by 2 mM ATP at peak RyR1 activation (Fig. 1a). Presence of 1 mM free $Mg^{2+}$ inhibited the channel at all $Ca^{2+}$ concentrations despite the presence of ATP or activating $Ca^{2+}$ conditions (Fig. 1a). In a competition experiment, $Mg^{2+}$ showed progressive inhibition starting at concentrations well below the physiological concentration of $Mg^{2+}$, with 50% lower ryanodine binding at a $Mg^{2+}/Ca^{2+}$ ratio of 2.2 (without ATP) and 3.5 (in the presence of 5 mM ATP) (Fig. 1b). The $IC_{50}$ was 111 μM for $Mg^{2+}$ and 174 μM for ATP/$Mg^{2+}$; $EC_{50}$ for ATP was 848 μM. In agreement with previous studies[26], ACP had a stronger activating effect on RyR1 than ATP and, under maximally activating $Ca^{2+}$, enabled RyR1 activity under conditions of $Mg^{2+}$ that normally fully inhibit the channel in the presence of ATP (Fig. 1c).

### Cryo-EM structure determination

Cryo-EM and single-particle analysis was carried out on RyR1 reconstituted in POPC-MSP1E3D1 nanodiscs at two different concentrations of free $Mg^{2+}$, ~1 mM and ~10 mM, and in the presence of AMP-PNP (ACP), a non-hydrolyzable ATP analog. We refer to such conditions as ACP/L$Mg^{2+}$ and ACP/H$Mg^{2+}$ henceforth. The overall resolutions for the RyR1-ACP/L$Mg^{2+}$ and RyR1-ACP/H$Mg^{2+}$ datasets were 4.4 Å and 3.1 Å, respectively, and increased locally using focused refinement. The symmetry-expanded focused map of RyR1-ACP/L$Mg^{2+}$ had a resolution of 3.8 Å in the channel's stem region formed by the central domain (CD), U-motif, transmembrane domain (TMD), and C-terminal domain (CTD) (residues 3668–5037). In RyR1-ACP/H$Mg^{2+}$, targeted refinement of the TMD and cytoplasmic assembly (CytA) yielded resolutions of 2.8 and 3.4 Å, respectively. Details on cryo-EM data collection, single-particle image processing, and model quality characteristics are summarized in Supplementary Figs. 1, 2, and Supplementary Table 1.

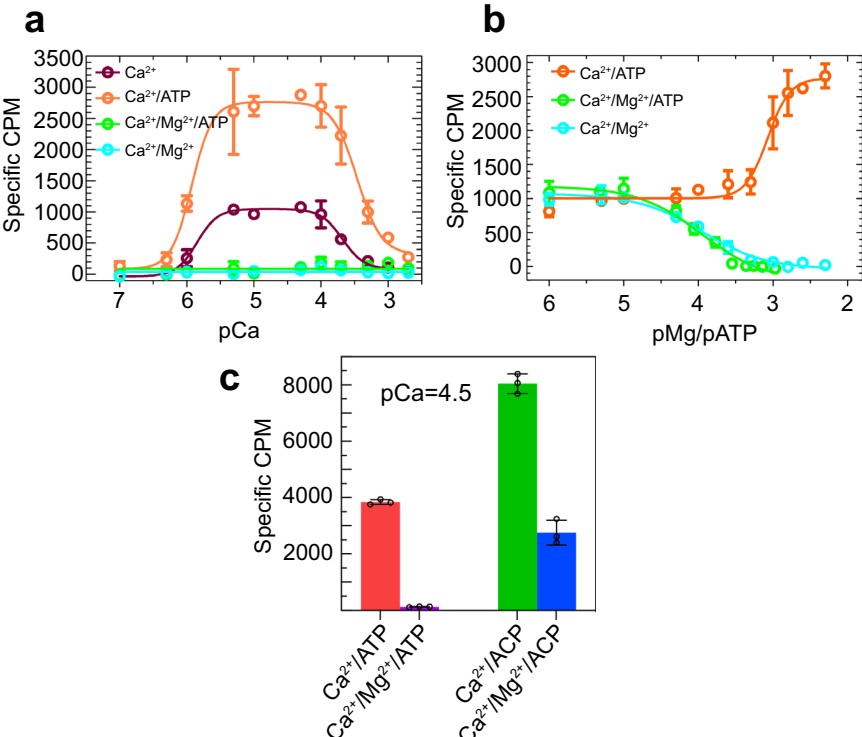

**Fig. 1 | Inhibition of RyR1 by $Mg^{2+}$. a** Ryanodine binding to rabbit skeletal SR membrane vesicles demonstrates a biphasic response of RyR1 over a wide range of $Ca^{2+}$ concentrations (100 nM–2 mM). At all $Ca^{2+}$ concentrations, ATP (2 mM) increased ryanodine binding, whereas $Mg^{2+}$ alone (1 mM) or $Mg^{2+}$/ATP (5 mM each) abolished channel activity. **b** Progressive inhibition of RyR1 by $Mg^{2+}$ (from 1 μM to 5 mM) in the presence or absence of equivalent concentrations of ATP, and

progressive activation of RyR1 by ATP (from 1 μM to 5 mM). $Ca^{2+}$ concentration was 50 μM for all data points. **c** At 50 μM $Ca^{2+}$, $Mg^{2+}$/ATP (5 mM each) abrogated RyR1 opening completely, while substitution of ATP by AMP-PCP resulted in further activation of the channel and incomplete inhibition by $Mg^{2+}$, respectively. Data represent mean±SEM specific [3H] ryanodine binding from three independent experiments. pCa = $-\log_{10}$ [$Ca^{2+}$], pMg = $-\log_{10}$ [$Mg^{2+}$], pATP = $-\log_{10}$ [ATP].

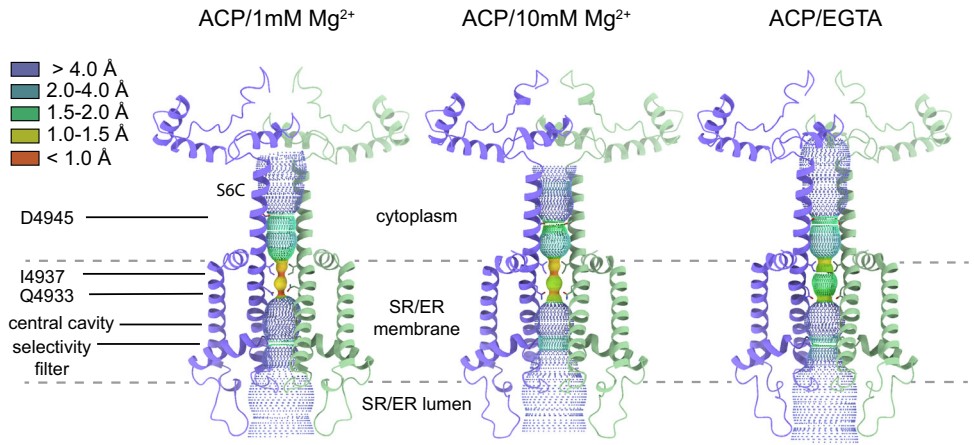

**Fig. 2 | Pore profiles of RyR1-ACP/Mg²⁺ and ACP/EGTA.** Protein-excluded dotted surfaces showing the ion permeation pathway and cytoplasmic extension of the S6 helical bundle (S6C) of RyR1 in the presence of ACP/Mg²⁺ (this work) and in the presence of EGTA (PDB ID: 7K0T). Landmarks include the hydrophobic (I4937) and polar gating residue (Q4933) at the ion gate and the negatively charged ring at D4945 in S6C. Residues 4821–5037 of two subunits in diagonal are displayed. The dashed lines indicate the approximate boundaries of the SR/ER membrane.

Except for the divalent cation and ACP concentration, the buffer conditions are identical to these we used in previous 3D reconstructions relevant for this study, determined either in the absence of divalent (i.e., in a mixture of EGTA and EDTA; henceforth RyR1-ACP/EGTA) or in the presence of 2 mM Ca²⁺ (which yielded an open and an inactivated conformation, henceforth referred to as RyR1-ACP/Ca²⁺-open and RyR1-ACP/Ca²⁺-inactivated, respectively)[26].

A defining characteristic of RyR1's conformation is the flexion angle[28]. This angle largely recapitulates the swiveling of the four quadrants of the top shell of the cytoplasmic assembly formed by the NTD, handle, SPRY, HD1, and P1 domains, which reflects structural transitions of the RyR1 upon its modulation by ligands[29] and becomes a determining factor in the classification of RyR cryo-EM images. The RyR1-ACP/LMg²⁺ 3D structure had a flexion angle of −1.5° (Supplementary Fig. 1), and further targeted 3D classification utilizing a quadrant-shaped mask in conjunction with symmetry expansion resulted in three main conformations of the cytoplasmic shell, with flexion angles of −0.4° (32%), −1.6° (24%), and −2.0° (33%). The RyR1-ACP/HMg²⁺ dataset had conformation homogeneity and exhibited a larger degree of swiveling with a flexion angle of −2.5°.

## Mg²⁺ lodges at the channel axis constricting the ion permeation pathway

The transmembrane domain of RyR1 has the typical 6-TM organization shared among several cation channels[29], where the sixth α-helix (S6, or inner helix) of each of the four subunits assembles as a helical bundle to form the ion permeation pathway. The C-terminal portion of S6 (S6C) extends the helical bundle into the cytoplasm. Pore profiles of RyR1 under ACP/LMg²⁺ and ACP/HMg²⁺ conditions revealed a closed pore with a radius narrowing to 1 Å at the hydrophobic gate I4937 (Fig. 2).

The ion permeation pathway revealed non-protein cryo-EM densities along the pore axis, a feature observed in other reconstructions of RyR. Notably, under the high Mg²⁺ condition, a density between the four side chains of D4945 of S6C became well defined, with the same or higher σ level than the protein density, and much higher than any central density in RyR1-ACP/EGTA (PDB ID: 7K0T) prepared under matching conditions except for the lack of Mg²⁺. As a comparison, an axial non-protein density in the selectivity filter region had a σ value of 1.75 that did not increase with the Mg²⁺ concentration. The density at position D4945 was substantially higher when compared to other reconstructions such as RyR1-ACP/Ca²⁺ (PDB ID: 5T15), or RyR1-ACP/

HCa²⁺ (PDB ID: 7TDG), while RyR1-ACP/LMg²⁺ showed intermediate density, altogether suggesting that the identity of this central density corresponds to Mg²⁺. Further supporting this assignment, in the absence of Mg²⁺ the side chain carbonyls of D4945 orient away from the fourfold axis, suggesting repulsion among them, whereas in the presence of Mg²⁺ the side chains of D4945 are oriented towards the central globular density (Fig. 3a).

In the absence of Mg²⁺ or Ca²⁺, there is a hydrogen bond between adjacent S6 helices, R4944-E4942′. In RyR1-ACP/LMg²⁺, the S6 helical bundle tightens, a trend that is enhanced in RyR1-ACP/HMg²⁺, where the pore further constricts forming a tripartite D4945′-R4944-D4938′ salt bridge that also connects to the putative Mg²⁺ mass at the level of D4945 (Fig. 3b, Supplementary Movie 1), also forming an additional E4942-H4832 interaction. When comparing different 3D reconstructions of RyR1 obtained with a variety of buffer conditions leading to a closed pore, the Mg²⁺ conditions resulted in the smallest constriction at D4945 (see Fig. 3c), providing further evidence that Mg²⁺ induces tighter association of the inner helices at a position slightly past the ion gate. As a reference, the S6 helix separation at I4937 (measured as distance between opposite $C_\alpha$) was 10.6 Å for RyR1-ACP/LMg²⁺ and 10.9 Å for RyR1-ACP/HMg²⁺, within the narrow range found for other closed structures (Fig. 3c). Thus, the I4937 hydrophobic gate displays a similar conformation under all closed-state conditions, while only the presence of Mg²⁺ induces further tightening of the cytoplasmic portion of the S6 helical bundle. Mg²⁺ occupancy of the D4945 site appears to hold the S6 helices more closely and to increase the resistance to separate during the closed-to-open transition.

## Stability of Mg²⁺ at the D4945 site

To elucidate the stability and dynamics of Mg²⁺ and Ca²⁺ at the central axis of the pore flanked by the carboxylate groups of D4945 from the four S6 helices, MD simulations of RyR1-ACP/LMg²⁺ (closed), RyR1-ACP/HMg²⁺ (closed), and RyR1-ACP/Ca²⁺ (open) were conducted in the presence of the divalent cations of interest, Ca²⁺ and Mg²⁺ (Supplementary Table 2). In the two RyR1 closed structures, the MD trajectories revealed that Mg²⁺ remained highly stable at the D4945 site, while Ca²⁺ left the site and moved toward the cytoplasm shortly after simulation was initiated (Fig. 4a, b, d, e)—a behavior consistently reproduced (see replicas in Supplementary Fig. 3). Consistent with this observation, the DFT-derived binding energies (ΔE_bind) in the two closed pore structures indicated a significantly higher affinity of the D4945 binding site for Mg²⁺ over Ca²⁺, as demonstrated by the greater

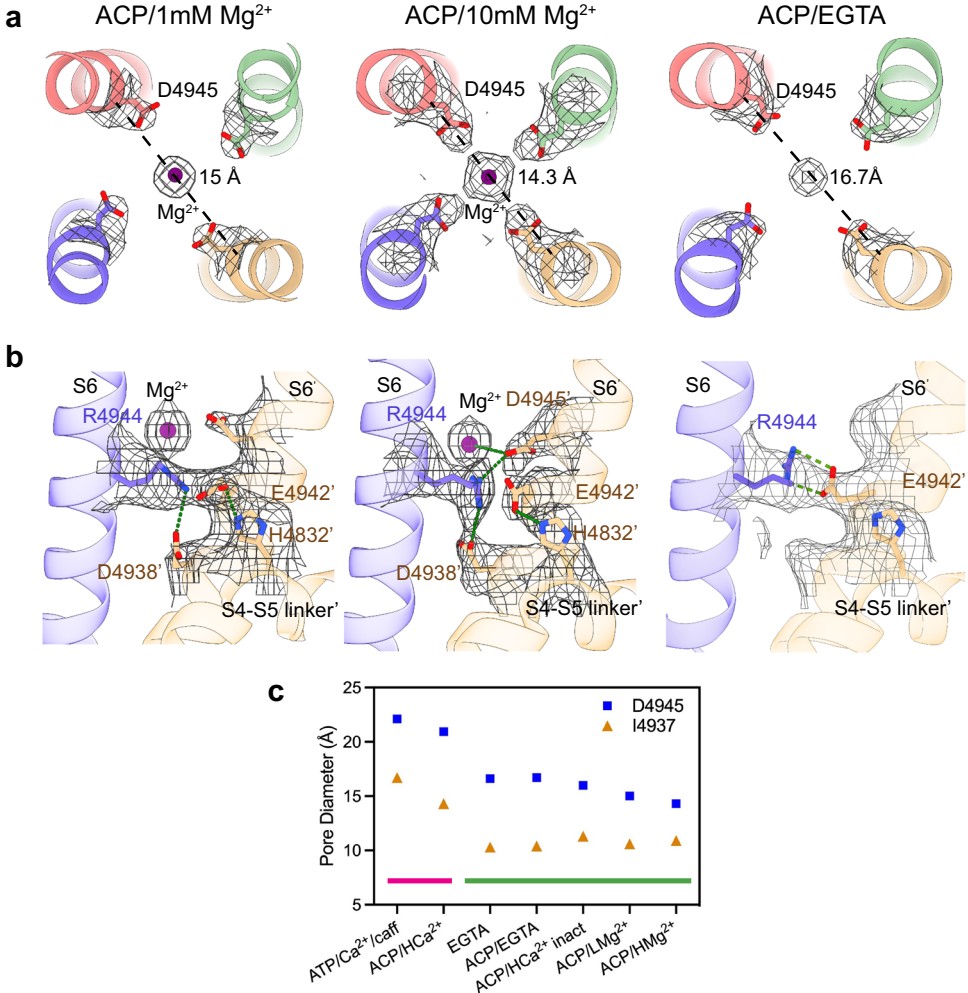

**Fig. 3 | Concentration-dependent effect of Mg²⁺ on the pore of RyR1. a** Fourfold cytoplasmic view of the S6C extension. The ring formed by D4945 shrinks by 1.7–2.4 Å in the presence of Mg²⁺ (purple). Pore diameter measured from the diagonal helices (Cα atom) is shown. Coulombic density around the putative Mg²⁺ site is displayed in mesh. Density attributed to Mg²⁺ in RyR1-ACP/10 mM Mg²⁺ can be seen up to 11 σ. **b** Mg²⁺ binding to the D4945 site forms a tripartite bridge between neighboring inner helices (S6) in the ACP/HMg²⁺ condition. The interaction network, D4945'-R4944-D4938', and E4942-H4832 extends from the centrally coordinated Mg²⁺ ion to the S4-S5 linker (via S6). The structures obtained at low Mg²⁺ or without Mg²⁺ lack such an interaction network. **c** Comparison of the pore diameter

(measured between opposite Cα of I4937 and D4945) of RyR1 under different conditions. Magenta line indicates open channels, green line indicates closed channels. The conditions and PDB IDs are as follows: ATP/30 μM Ca²⁺/caffeine - open (5TAL), ACP/2 mM Ca²⁺ - open (7TDH), metal-free - closed (EGTA; 5TB0), ACP/EGTA - closed (7K0T), ACP/2 mM Ca²⁺ - closed inactivated (7TDG), ACP/1 mM Mg²⁺ - closed (7K0S; this work), ACP/10 mM free Mg²⁺ - closed (7UMZ; this work). The four conditions on the right as well as ACP/2 mM Ca²⁺ - open correspond to channels prepared in identical buffer conditions except for the divalent metal. The cytoplasmic portion of the S6 bundle is narrowest in the presence of Mg²⁺.

values for the Mg²⁺-D4945₄ complex compared to the Ca²⁺-D4945₄ complex (Supplementary Table 3). Furthermore, the binding energy of the Mg²⁺-D4945₄ complex was greater in the structure determined at high Mg²⁺ concentration compared to that determined at low Mg²⁺ concentration, aligning with the MD simulations where Mg²⁺ in the RyR1-ACP/LMg²⁺ system exhibited greater ion displacement fluctuation than Mg²⁺ in the RyR1-ACP/HMg²⁺ system. This suggests that the conformation of RyR1 at the higher Mg²⁺ concentration results in allosteric effects that induce a configuration of the pore (and of D4945) with higher affinity for Mg²⁺.

In the closed states, MD simulations illustrated that the carboxyl oxygens of D4945 engaged in hydrogen bond interactions with first-shell water molecules, establishing an indirect interaction between the cation and the protein's residues (Supplementary Movies 2, 3, Fig. 5a). This arrangement implies that D4945 may serve as a second-shell ligand, contributing to the stabilization of Mg²⁺. The role of aspartate residues serving as second-shell ligands for divalent cations is a common observation in protein structures[30]. A hydrogen bond network

involving D4945', R4944, D4938', and Mg²⁺ was observed in the RyR1/HMg²⁺+Mg²⁺ model (Supplementary Movie 3, Fig. 5b), and corresponds to the network independently identified by cryo-EM (Fig. 3b, Supplementary Movie 1), highlighting the tighter packing of the cytoplasmic part of S6 helix and supporting the presence of Mg²⁺ at the D4945 site. All the above MD and DFT analyses emphasize the important role that Mg²⁺ plays in the resting state of the channel and likely explains the malfunction of RyR1 in the presence of anomalous Mg²⁺ levels[8,9]. In the open conformation, D4945 did not retain Mg²⁺ or Ca²⁺ (Fig. 4c, f), consistent with their known permeability[6,31]. Thus, our MD results indicate that in the open state, the D4945 site is permeable, allowing the passage of both Mg²⁺ and Ca²⁺ ions through the channel without retention.

## Mg²⁺ binds to the high-affinity Ca²⁺ site without triggering activation

The Ca²⁺ activation site, at the CD/CTD interface[17], was empty and maximally open in the RyR1 cryo-EM map determined at low Mg²⁺, with

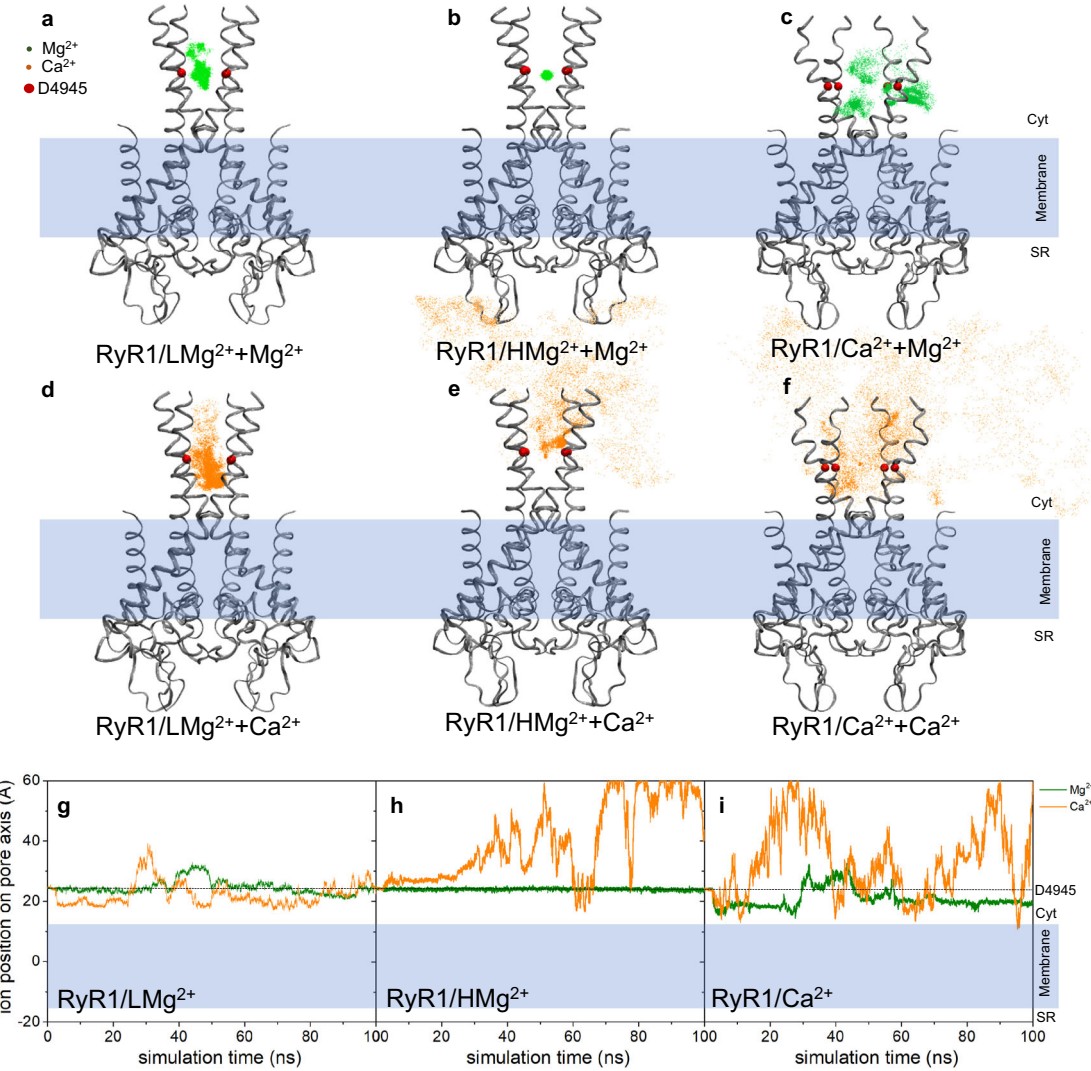

**Fig. 4 | Molecular dynamics of the pore domain. a–f** Displacement of $Mg^{2+}$ and $Ca^{2+}$ ions is depicted with green and orange dots, respectively, at the D4945 site (red spheres) during 100 ns MD simulations. Each dot represents the position of the indicated cation through the collection of MD snapshots taken at intervals of 0.02 ns. The ribbon structures of the RyR1 pore domain in various conformations are illustrated in their initial configurations. **g–i** Ion displacement in relation to the z-axis of the channel over the course of simulation time.

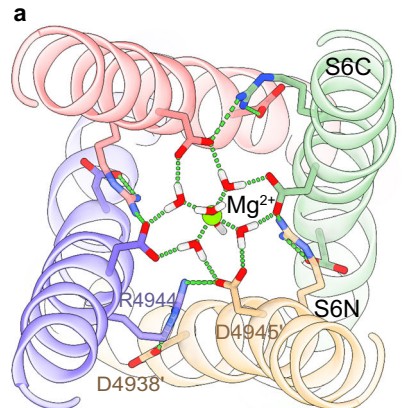

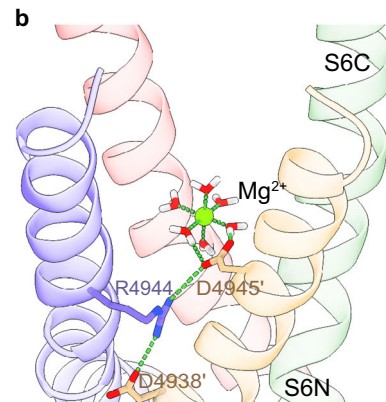

**Fig. 5 | Molecular dynamics snapshot of RyR1/HMg$^{2+}$ + Mg$^{2+}$. a** Cytoplasmic view of the S6 helical bundle illustrating the involvement of the D4945 carboxyl oxygens in hydrogen bond interactions with first-shell water molecules, leading to stabilization of $Mg^{2+}$ (green sphere) at this site. **b** Side view showcasing the hydrogen bond network involving D4945′, R4944, D4938′, and $Mg^{2+}$.

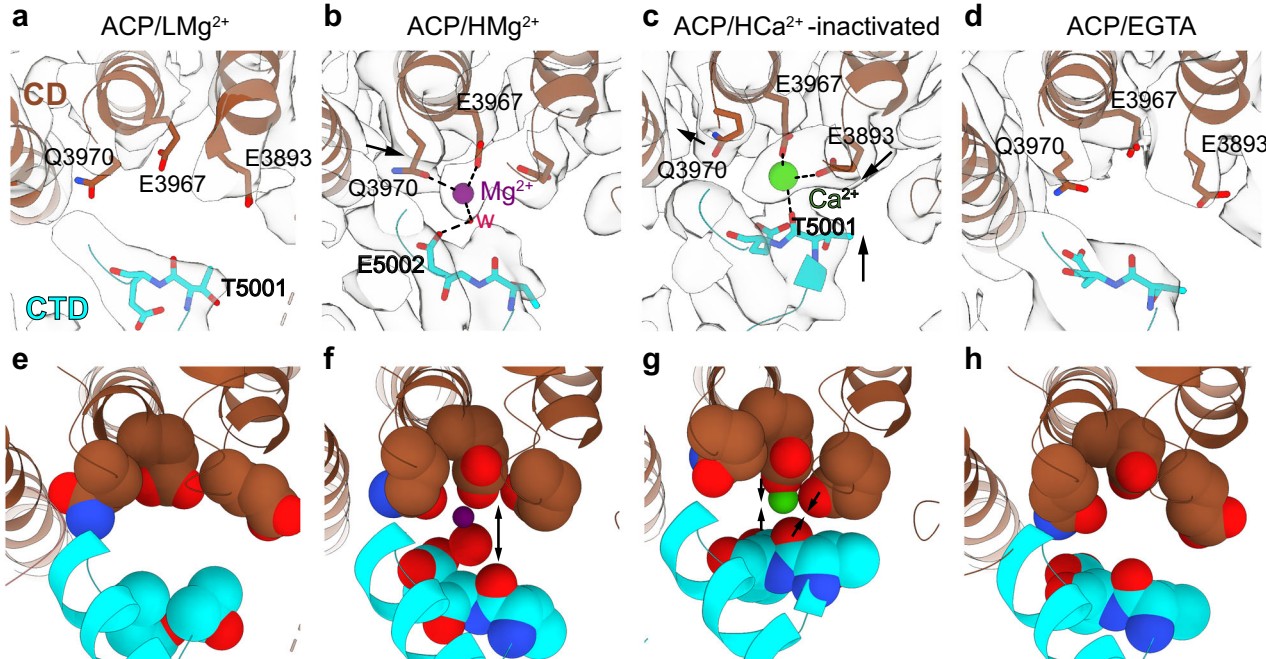

**Fig. 6 | Binding of Mg²⁺ at the high-affinity Ca²⁺ activation site. a–d** Cryo-EM of the Ca²⁺ binding site at the CD/CTD interface reveals a shared binding site for Ca²⁺ and Mg²⁺. At low Mg²⁺ concentration, the site is empty. At high Mg²⁺ concentration, the site is occupied by partially hydrated Mg²⁺, which imposes a different configuration compared to Ca²⁺. In contrast to Ca²⁺ binding to E3893, E3967, and the carbonyl of T5001, Mg²⁺ binds to E3967, Q3970, and a water molecule connected to E5002 (distances < 3 Å), but it is separated from E3893 (4.7 Å). Coulombic densities surrounding the divalent binding site are depicted. RyR1 without Ca²⁺ (PDB ID: 7K0T) or Ca²⁺-inactivated RyR1 (PDB ID: 7TDG) are shown for comparison; see Supplementary Fig. 4 for RyR1 in the open state. Arrows indicate the conformational changes effected by Mg²⁺ or Ca²⁺. Pore axis is on the left. **e–h** Space-filling representation of the Ca²⁺ activation site under conditions corresponding to the above panels. Mg²⁺ and its water of hydration cause slight expansion of the cavity, indicated by the separation of T5001-E3967 by 1 Å and lack of interaction with E3893 in RyR1-ACP/HMg²⁺. In contrast, in the Ca²⁺-inactivated structure, Ca²⁺ gravitates towards E3893, releasing Q3970, and the T5001-E3893 contact shrinks by ~2 Å closing the cavity on the E3893 side.

a conformation almost identical to that of RyR1-ACP/EGTA (Fig. 6a, d, e, h). At 10-fold higher free Mg²⁺ concentration and in the absence of Ca²⁺, a clear density visible up to 7 σ appeared within the Ca²⁺ activation site (CAS), which indicates binding of Mg²⁺ (Fig. 6b). Mg²⁺ interacted with E3967 and Q3970 (from the CD) within its first coordination shell, and with E5002 (from the CTD) through its hydration shell. When considering the angle between the line passing through the Ca²⁺-binding residues of the CD (E3893, E3967, and Q3970) and the axis of the closest CTD α-helix (residues 5005–5016), the angle decreased by ~2°. This is a small "pull" when compared to the ~15° angle closure produced by Ca²⁺ activation (measured in two open states) (PDB ID: 5TAL[17] and PDB ID: 7TDH[26]; Supplementary Fig. 4), in agreement with the inability of Mg²⁺ to activate the channel. Comparison of the RyR1 structures obtained in the presence of Ca²⁺ and HMg²⁺ revealed that both divalent cations interact with E3967, but Mg²⁺ interacts with Q3970 instead of E3893 and with E5002 instead of the carbonyl of T5001 (Fig. 6b, c, Supplementary Fig. 4, Supplementary Movie 4). Thus, the CD pivots either leaving the cavity more open on its N′ terminus in the presence of Mg²⁺ or more open on its C′ terminus in the presence of Ca²⁺. Overall, mainly through side chain reorientation of the CD and CTD residues, Mg²⁺'s coordination stabilizes the interaction between the CD and CTD domains, but the partly hydrated Mg²⁺ appears to act as a spacer between them, pushing the CTD down and stabilizing the CAS in the position characteristic of the closed state.

## Conformational change of the EF-hand domain of RyR1 at high Mg²⁺

The EF-hand domain, spanning amino acids 4071–4131, forms a distinctive globular shape protruding from the cytoplasmic assembly via its two flanking helices: the incoming α-helix emerges from the CD, and the outgoing α-helix goes to the U-motif. The EF-hand domain has two helix-loop-helix modules, where the EF1 and EF2 loops have negative charges that can bind Ca²⁺ with low affinity[32]. At low Mg²⁺, the position of the EF-hand domain of RyR1 resembles that observed in the presence of EGTA. However, at high Mg²⁺ concentration, the EF-hand rotated towards the 122-residue-long cytoplasmic loop between the S2 and S3 transmembrane helices of the neighboring subunit. Specifically, residues E4075 and K4101 from the EF-hand domain were within salt-bridge distance of residues R4736 and D4730 of the S2–S3 loop, respectively (Fig. 7a, ACP/HMg²⁺, Fig. 7b). This configuration is almost identical to that found for Ca²⁺-inactivated RyR1 (Fig. 7a, ACP/HCa²⁺ inactivated)[26]. RyR1 in the open conformation[17,26] also shows approximation of the EF-hand domain and S2–S3 loop, but the two domains are more separated (Fig. 7a, ACP/HCa²⁺ open and ACP/LCa²⁺/caffeine open) than in the inactivated conformation. Ca²⁺ was not resolved within the EF-hand domain of Ca²⁺-inactivated RyR1 or any other RyR1 structure in the presence of Ca²⁺, and no coordinated metal is resolved at high Mg²⁺ either, largely due to the low resolution of this domain that hampers precise modeling of the main chain or visualization of any metal bound. Potential reasons for the low resolution of the EF-hand domain could be fractional occupancy of the site by cation(s) leading to the coexistence of conformations, and that this domain protrudes from the cytoplasmic domain allowing higher mobility.

Another possible explanation for the conformational change observed in the EF-hand domain is that binding of divalent cations to the CAS in the CD domain could induce repositioning of the EF-hands, since the CD and EF-hand domains are connected (Fig. 7c). However, given that the translation of the EF-hand domain (~10 Å) is much larger than the translation of the connected domains (less than 2 Å for the CD and U-motif) and that the EF-hand domain only moves in the presence of Ca²⁺ or high Mg²⁺, metal chelation seems the major mechanism to account for the reorientation of the EF-hand domain. In the canonical

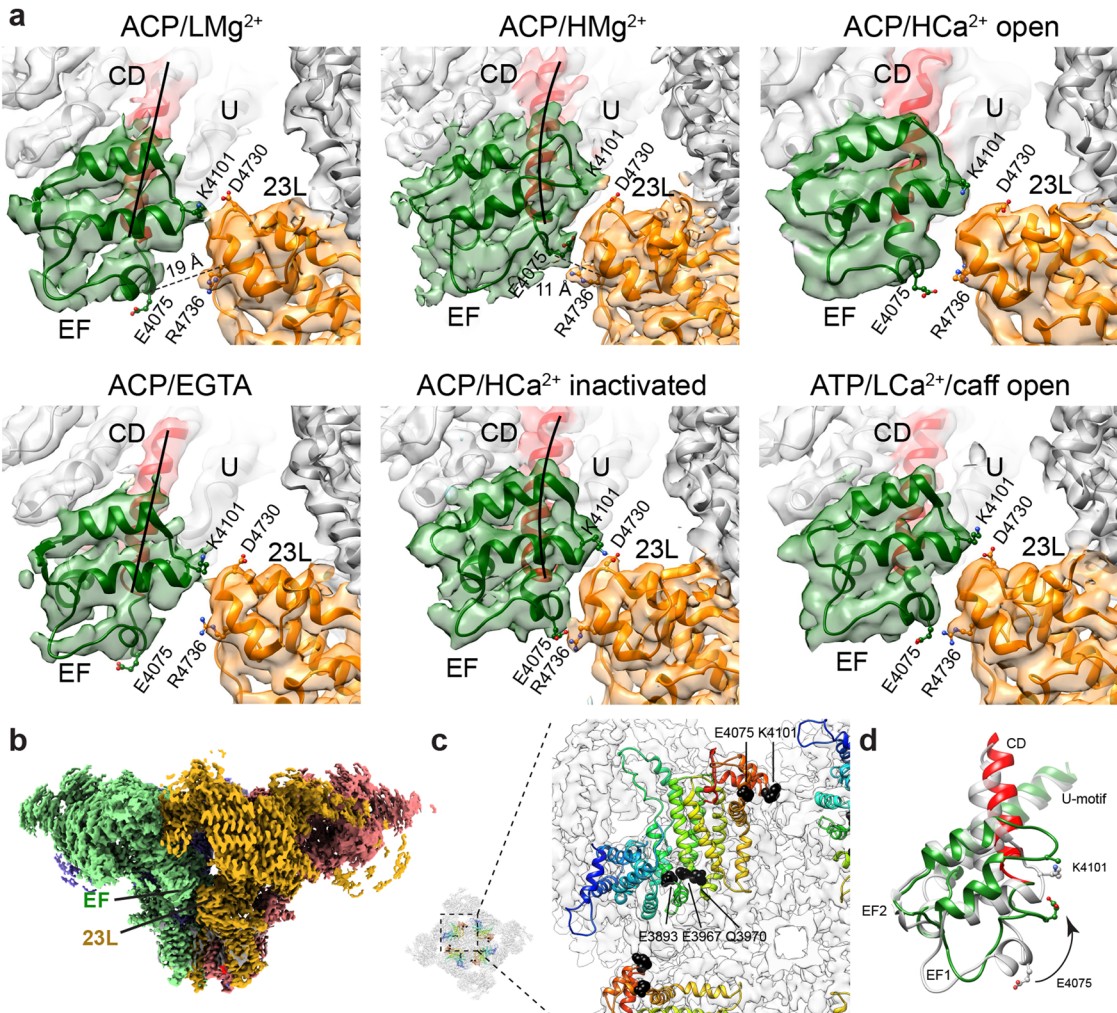

**Fig. 7 | Mg²⁺-induced conformational changes at the EF-hand domain in the context of closed, inactivated, and open channels. a** Transition from the ACP/LMg²⁺ to the ACP/HMg²⁺ condition results in a conformational change in the EF-hand domain (green). A rotation of the domain concomitant with a tilt in the incoming alpha helix (residues 4053–4070 from the CD, in red, with helical axis indicated with a black line) approximates E4075 and R4736 from -19 Å to -11 Å (measured at the Cα). This orientation enables the formation of two salt bridges with the S2–S3 cytoplasmic extension (23L; orange) of the neighboring subunit. Transition from ACP/EGTA to ACP/HCa²⁺-inactivated conditions undergoes an analogous conformational change (each shown below for comparison). In open RyR1s, induced either by 2 mM Ca²⁺ or by 30 µM Ca²⁺ plus ATP and caffeine, the EF-hand domain moves less than in the HCa²⁺ or HMg²⁺ conditions. **b** Side view of RyR1 with one EF-hand domain and S2–S3 cytoplasmic loop facing the viewer. Each

subunit is displayed in a different color. The most proximal part of the cytoplasmic shell is removed for clarity. **c** Relative position of the EF-hands and CD seen from the TMD side. The polypeptide chain is color-coded from blue at the CD N' terminus to red at the EF-hand C' terminus. The three CAS residues in the CD and the two salt-bridge-forming EF-hand residues are indicated. Proximity between the CAS and the EF-hand domain suggests correlated movements (see also Supplementary Movie 4). However, the EF-hand domain moves more than the CD, indicating ability to move independently, and appears to pull away the C' terminal helix of the CD (in red in **a** and **d**) from the rest of the CD. **d** Conformational change of the EF-hand upon increasing the free Mg²⁺ concentration from 1 mM (gray) to 10 mM (red and green as in **a**). Coordination of Mg²⁺ by the negative charges of the EF loops likely underlies the conformational change.

mechanism of Ca²⁺ sensing, chelation of divalent cations by the helix-loop-helix motifs of the EF-hands changes the angle between such helices, which is also obvious in RyR1's EF-hands (Fig. 7d). Reorientation of RyR1's EF-hands at high Ca²⁺ or Mg²⁺ also resulted in detachment of the incoming α-helix (residues 4053–4070) from the CD (see Fig. 7a and Supplementary Movie 4), further corroborating that the conformational change in the EF-hand domain forced reorientation of its N' terminus with respect to its C' terminus. The immobility of the outgoing α-helix suggests higher rigidity of the U-motif.

Our results indicate that the EF-hand domain of RyR1 can sense both Ca²⁺ and Mg²⁺. Considering that micromolar Ca²⁺ induced more movement in the EF-hand than 1 mM Mg²⁺ (compare Fig. 7a, ACP/LMg²⁺, and ATP/LCa²⁺/caffeine) reflects a higher affinity of the EF-hand domain for Ca²⁺ than for Mg²⁺. We propose that this site, shared by both Mg²⁺ and Ca²⁺, may correspond to a hypothesized inhibitory

divalent sensor, where activation of this sensor adds an additional interaction between neighboring subunits (Fig. 7b).

## Mg²⁺ lodges within the ATP binding site

ATP binds to a pocket formed by the U-motif, S6C, and CTD within the same subunit[17]. The S6C and CTD constitute the innermost portion of the binding pocket that houses the nucleotide base moiety, whilst the U-motif contributes binding to the distal triphosphate tail (Fig. 8a). Presence of Mg²⁺ resulted in a conspicuous larger mass around the ACP triphosphate moiety, which was well resolved in the RyR1-ACP/HMg²⁺ map. Two putative densities could be attributed to Mg²⁺; one of them forms a bidentate complex with oxygens in the α and γ phosphate, and the other connects with the oxygens of the phosphate backbone (Fig. 8b). Overall, Mg²⁺ triggered clockwise in-plane rotation (as seen from the cytoplasmic side) of the S6C-U-motif region, drawing the ACP

**a**

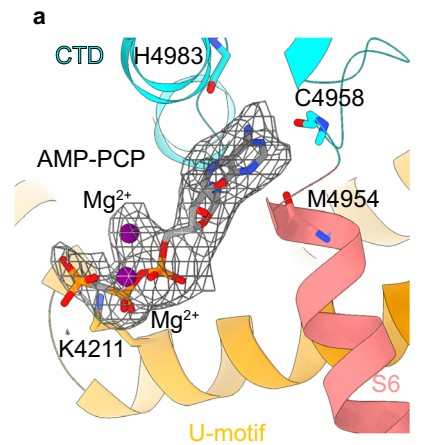

**b**

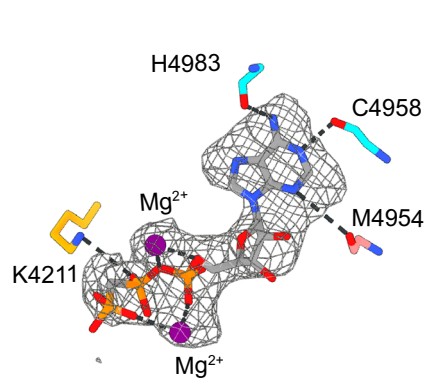

**Fig. 8 | Mg²⁺ forms a stable complex with ACP at the U-motif-S6-CTD- interface. a** An ACP (AMP-PCP) molecule bound to the U-motif, S6C, and CTD domain interface contains two additionally coordinated Mg²⁺ ions clearly visible in the ACP/HMg²⁺ condition. **b** Densities attributed to two Mg²⁺ ions and ACP in the ACP/HMg²⁺

cryo-EM focused map at 2.8 Å resolution. The Mg²⁺ at the bottom adopts a bidentate geometry with the terminal oxygens of alpha and gamma phosphates, whereas the Mg²⁺ at the top coordinates with the oxygens from the ACP backbone.

molecule into the cavity in a concentration-dependent manner. The U-motif, which is close to the phosphate tail in RyR1-ACP and RyR1-ACP/LMg²⁺ due to interactions via K4211, K4214, and R4215, is separated (by 10 Å) from the phosphate tail and forms an extra contact with the CTD (H4983) in RyR1-ACP/HMg²⁺, similar to the Ca²⁺-inactivated structure that stabilizes an inward-drawn conformation of the nucleotide base. It appears that the nucleotide settling deeper into its binding pocket is more a consequence of the conformational rearrangement caused by coordination of Mg²⁺ at other sites of the channel than a direct consequence of Mg²⁺ binding to the nucleotide.

### The lipid-binding pocket in the TMD is occupied

The domain-swapped inter-subunit space of RyR1-ACP/HMg²⁺, formed by S3/S4 and S5/S6 of the neighboring subunit, was occupied by non-protein densities identified as two molecules of phosphatidylcholine (PC16:0/18:1), based on the addition of this lipid during reconstitution into nanodiscs and the cryo-EM densities for their head group (Supplementary Fig. 5). Lipid-1 primarily interacts with six amino acids at S5/S6, while lipid-2 interacts with twelve amino acids at S3/S4 (Fig. 9). Together, the lipid buries 783 Å² of the cavity lined by hydrophobic side chains of S3, S4, S5, and S6. The configuration of the lipid moiety and surrounding pocket is practically identical to that found in closed RyR1 structures solved in the presence of lipids, e.g., RyR1-ACP/Ca²⁺-inactivated (PDB ID: 7TDG[26]), Y522S RyR1 closed (PDB ID: 7T64[33]), RyR1-FKBP bound closed (PDB ID: 7TZC[34]), as well as closed RyR2 R176Q (PDB ID: 6WOU[25]), with more of the head and fatty acyl tail of lipid-2 resolved in RyR1-ACP/HMg²⁺. As mentioned earlier, lipid-2 forms a steric clash with S4 of open channels[26], which supports the participation of lipids in RyR's gating transitions.

### Discussion

We carried out functional, structural, and molecular dynamics studies to understand RyR1 inhibition by Mg²⁺. The structural analysis of pure RyR1 particles in the vitrified state was obtained in the presence of the cytosolic modulators Mg²⁺ and an ATP analog. The use of nanodiscs to preserve the transmembrane domains in a detergent-free manner provided a lipidic environment with millimolar concentrations of Mg²⁺ and ATP as representative as possible of the native resting closed state of the RyR1. In a second reconstruction, we used a 10-fold free Mg²⁺ concentration range to enable higher binding site saturation and help resolve ion occupancy. Using high-resolution cryo-EM and MD, we identified putative Mg²⁺ binding sites and characterized the effect of

Mg²⁺ on RyR1's conformation. Based on the observations reported here, we propose a multi-factorial mechanism of inhibition by Mg²⁺.

The main inhibitory effect of Mg²⁺ derives from its binding to the pore by coordinating the four inner helices along the ion permeation pathway. This coordination site is provided by the four D4945 residues in the cytoplasmic portion of S6. The side chains of D4945 extend towards the center, in contrast to RyR1 without Mg²⁺ where such side chains are tucked (see Fig. 3). At the center of the extended side chains there is a self-standing mass attributed to Mg²⁺, which increases in density at the higher Mg²⁺ concentration. MD and DFT calculations validated that D4945 in the inner helices can accommodate Mg²⁺ but not Ca²⁺ and that coordination with Mg²⁺ occurs exclusively in the closed conformation (Fig. 4a, b). At 10 mM free Mg²⁺, the axial density attributed to Mg²⁺ near D4945 is well defined, with a density of 6 σ matching that of surrounding residues. MD substantiates that the side chains of D4945 engage in hydrogen bonds with the first shell of hydration of Mg²⁺ (Fig. 5). Thus, our results from cryo-EM and MD together suggest that D4945 in the inner helices constitute a binding site for Mg²⁺ along the closed ion pathway. Coordination of Mg²⁺ to the inner helices was less favorable in the open conformation (Fig. 4c), supporting the fact that Mg²⁺ permeates the activated channel and plays a role in inhibition by competing for ion flow, in agreement with functional data[6,21,23].

An obvious difference between the 3D reconstructions of RyR1-ACP/Mg²⁺ and RyR1-ACP/EGTA is an extra density in the vicinity of the known ATP binding site (Fig. 8). This implies that under physiological conditions, it is Mg²⁺-ATP (not ATP alone) that binds within the U-motif/S6/CTD cavity. Mg²⁺ likely increases the structural continuity between domains by filling up the cavity. Considering that ATP is indispensable for skeletal-type EC coupling[8], reducing flexibility at this interface probably increases the efficiency in the long-range allosteric relay of the signal from the DHPR to RyR1's ion gate. Because ATP is an activator and Mg²⁺ enhances the ATP interaction with RyR1, it appears that as suggested earlier, ATP exerts its activating effect on the channel in complex with Mg²⁺[7].

The cryo-EM density map shows that Mg²⁺ occupies the high-affinity Ca²⁺ activation site in a concentration-dependent manner. At 1 mM free Mg²⁺, the high-affinity Ca²⁺ binding site is empty, while at 10 mM free Mg²⁺ the cation lodged into the high-affinity Ca²⁺ binding site (Fig. 6a, b). The ionic strength used in the cryo-EM experiments (0.635 M KCl) was higher than physiological, decreasing the affinity of Mg²⁺ to this site[22]. Thus, most likely, at physiological ionic strength, the

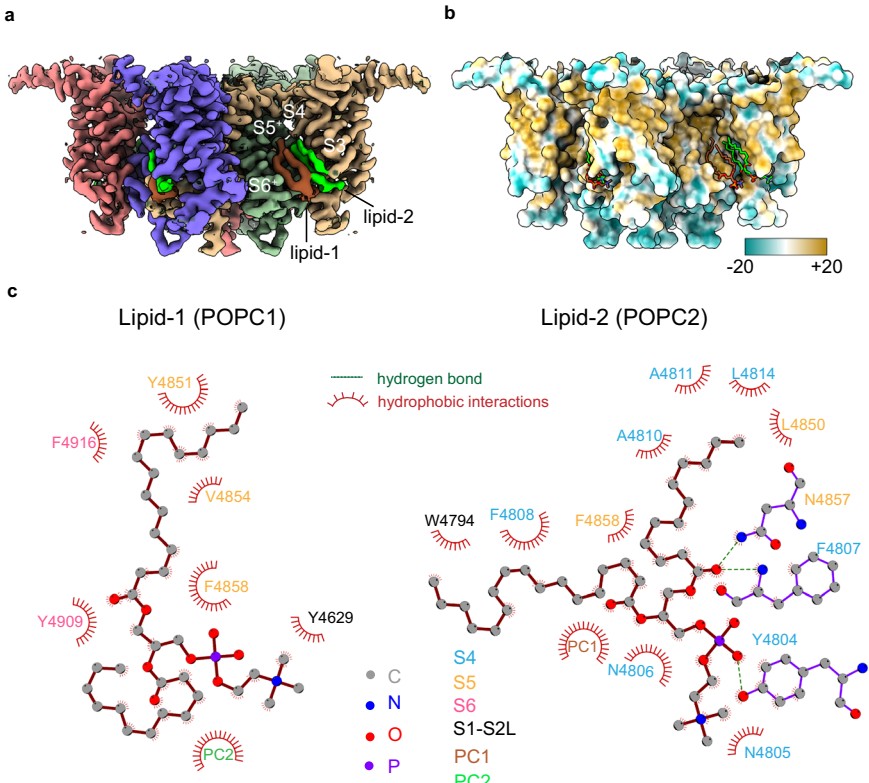

**Fig. 9 | A lipophilic crevice of RyR1 binds two lipids in the presence of ACP/HMg²⁺. a** Density for the transmembrane domain and bound lipids in the 2.8 Å resolution cryo-EM map focused on the central and transmembrane domains. The two partial lipids are part of two PC16:0/18:1 molecules that bind to a domain-swapped crevice formed by S3/S4 and S5/S6 helices of neighboring subunits. **b** Molecular lipophilicity potential map of the transmembrane domain showing a hydrophobic pocket where two lipids are bound. Hydrophilic and hydrophobic amino acids are colored cyan and yellow, respectively. **c** The two lipids are stabilized by contacts with S5/S6, S1-S2 loop (POPC1; left), and S4/S5 (POPC2; right), mediated by their hydrophilic head and hydrophobic tail. Hydrogen bond and hydrophobic contacts shown are <3.4 Å and <3.9 Å respectively.

CAS site should be at least partially occupied by 1 mM Mg²⁺. Even if both cations compete for the same site, owing to their different coordination properties they induce different effects in the CD/CTD interaction: Ca²⁺ binding induces closure of the CAS cavity (i.e., approximation of the CD and CTD domains; Supplementary Fig. 4) which leads to opening of the ion gate[17], while the larger hydrated Mg²⁺ keeps the CAS cavity in a moderately open configuration, similar to that adopted in the absence of Ca²⁺ (RyR1-ACP/EGTA) (Fig. 6d) where the ion gate is closed. Thus, the presence of Mg²⁺ at the cavity stabilizes an orientation of the CTD that keeps the inner helices of RyR1 in the closed state.

It was suggested that one of the mechanisms of Mg²⁺ inhibition could be competition for the high-affinity Ca²⁺ binding site[7,8,20,21]. Ca²⁺ occupancy of its high-affinity cavity is in the μM range, whereas Mg²⁺'s is in the mM range; implying at least a 1000-fold higher affinity of Ca²⁺ versus Mg²⁺ for the site. This relative affinity between Mg²⁺ and Ca²⁺ corresponds approximately to the relative free concentrations of the two cations in the cytoplasm of skeletal muscle during activation, revealing a fine-tuning of the CAS site to the ratio between both divalent cations in the cytoplasm and supporting an inhibitory role of Mg²⁺ at rest (i.e., sub μM free Ca²⁺ concentration). The antagonistic action of Mg²⁺ would be easily overcome as cytoplasmic Ca²⁺ increases. Mg²⁺ occupancy at this site can also explain the inability to trigger EC coupling at 10 mM free Mg²⁺, concentration at which Ca²⁺ competition may be unfavorable[8,9].

We described an inactivated conformation of RyR1 at high (inactivating) Ca²⁺ concentration, with Ca²⁺ bound to the CAS while the pore is closed[26]. There is a noticeable difference between the Mg²⁺-inhibited

and the 2 mM Ca²⁺-inactivated state. Compared to the tightly closed CAS cavity in the 2 mM Ca²⁺- inactivated state, the CAS cavity is open when occupied at high Mg²⁺ concentration (see Fig. 6f, g), which could have the opposite function, to facilitate exchange with Ca²⁺ by keeping the CD/CTD domains separated. Thus, high Ca²⁺ inactivates the channel preventing further activation, while Mg²⁺ inhibits the channel leading to a conformation of the CAS site more ready to be activated.

In the presence of divalent cations, the EF-hand domain of RyR1 moves towards the cytoplasmic S2–S3 loop of the neighboring subunit (Fig. 7). While the conformational change induced by CAS occupancy in the close vicinity may help positioning the EF-hand domain, the movement of the EF-hand is of larger magnitude, suggesting direct interaction of the cation at the EF-hand loop(s). This was well documented for the activated conformation induced by μM Ca²⁺, showing rotation and translation of the EF-hand with respect to the CD (9.7° and 6.2 Å versus 2.9° and 1.3 Å, respectively)[33]. In the inactivated conformation induced by mM Ca²⁺ the EF-hand domain moved further (Fig. 7a), with two salt bridges between the two domains joining two adjacent RyR1 subunits and presumably stabilizing the non-conductive conformation of the pore[26]. The important role of the two putative salt bridges was supported by three human disease mutations characterized by inactivation-deficient channels[35], where these mutations would hamper the interaction between the EF-hand and the S2–S3 loop. The Ca²⁺-sensing ability of the EF-hand domain of RyR1 was also shown before, where scrambling the negatively charged residues within the EF1 loop reduced the sensitivity of the channel to Ca²⁺-induced inactivation[36]. In the presence of 10 mM free Mg²⁺ the EF-hand domain of RyR1 moved towards the cytoplasmic S2–S3 loop of the neighboring

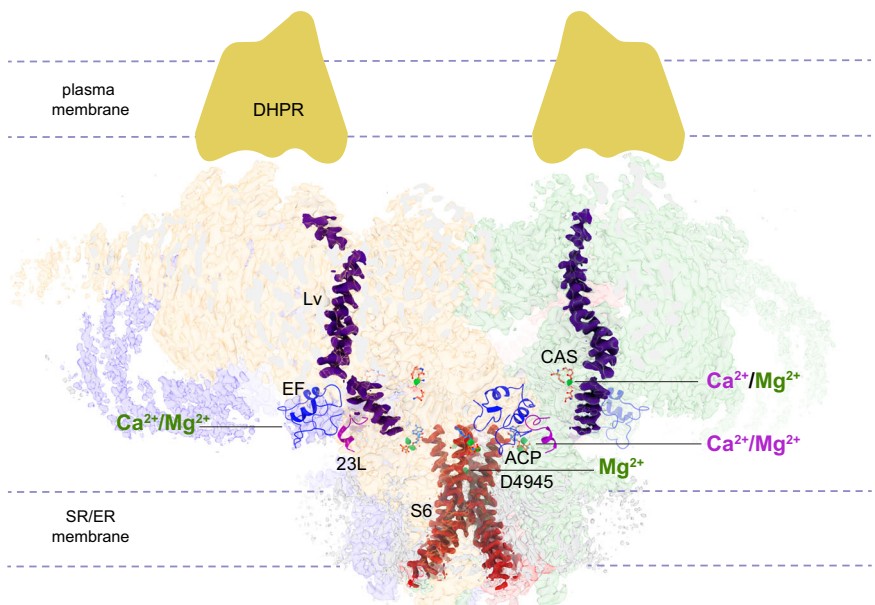

**Fig. 10 | Multiple Mg²⁺ and Ca²⁺ binding sites within RyR1.** A central slice of the RyR1 cryo-EM density (side view) is shown. At each site, the divalent cations act as activators (magenta font) or as inhibitors (green font). The high-affinity Ca²⁺ activation site (CAS) exhibits a lower affinity for Mg²⁺, which exerts an inhibitory effect. The EF-hands are proposed to function as a low affinity, inhibitory Ca²⁺/Mg²⁺ module that binds to the S2−S3 cytoplasmic loop (23L; partially shown for clarity) of the neighboring subunit. Binding of either Ca²⁺ or Mg²⁺ to the ATP site is likely to enhance structural continuity between its neighboring domains (U-motif, S6, and CTD) and produce an activating effect. Binding of Mg²⁺ to the D4945 site at the fourfold axis tightens the S6 helical bundle stabilizing the closed state, while no binding of Ca²⁺ is observed at this site. We hypothesize that four long levers (Lv; only two shown for clarity) provide a long-range allosteric pathway conveying the excitatory input from the DHPR to RyR1's stem region. Abatement of the levers induced by conformational coupling, or by binding of Ca²⁺ to the CAS, is expected to trigger the separation of S6, resulting in the removal of Mg²⁺ coordination at the D4945 site. Mg²⁺ ions are shown in green.

subunit almost to the same extent than in the presence of 2 mM Ca²⁺ (Fig. 7a), suggesting that the EF-hand domain may correspond to a predicted low-affinity inhibitory site shared between Ca²⁺ and Mg²⁺[7,16,20].

As Mg²⁺ is always present in the cell, it necessitates an explanation of how its inhibition is released. In skeletal muscle the T-tubules, invaginations of the plasma membrane, bring the voltage-gated L-type Ca²⁺ channel (Ca_V1.1 or DHPR) into very close contact with the RyR1, making for efficient conformational coupling between the two proteins and bringing about RyR1-mediated Ca²⁺ release and muscle contraction as a response to a nervous impulse[37–39]. Studies of the influence of Mg²⁺ showed a strong effect of its concentration on EC coupling in skinned skeletal muscle fibers. At 1 mM free Mg²⁺, the muscle fibers were quiescent, and only DHPR activation could overcome Mg²⁺ inhibition, leading to RyR1 channel opening[8,9]. Lowering the concentration of Mg²⁺ 10-fold triggered massive spontaneous Ca²⁺ release, while a 10-fold increase of Mg²⁺ abrogated depolarization-induced Ca²⁺ release[8,9,14]. This, among other findings, led to the hypothesis that DHPR activates the RyR1 by removing the potent inhibitory effect of Mg²⁺ at its physiological concentration of 1 mM[13]. In other words, activation of the DHPR would decrease the affinity of RyR1 for Mg²⁺, which would account for Ca²⁺ release upon depolarization.

Our structural work mirrors these functional findings and provides strong structural basis for the DHPR control of Mg²⁺ inhibition. We suggest that unlocking the S6 helical bundle is triggered by conformational coupling to DHPR, which would then abrogate the effect of the electrostatic network coordinating Mg²⁺. This leaves the question of how the DHPR effects such change in S6. One potential mechanism is via a ~100 Å-long lever that spans from the DHPR-facing region of RyR1 to the base of the CD (Fig. 10) that abates its angle by 6° when RyR1 is stimulated with Ca²⁺[33]. It is likely that this lever forms part of the long-range allosteric pathway connecting DHPR and RyR1's channel, as a severe RyR1 disease mutation in the DHPR-facing region of the lever abated the angle of this lever by 5° under closed-state conditions[33] and increased the sensitivity of DHPR to depolarization by 40 mV[40]. Thus, the change in the DHPR voltage sensor upon depolarization, potentially moving (abating) the long levers of RyR1, could constitute a mechanism to directly induce a separation of the inner helices, disrupting the Mg²⁺ coordination and unblocking the channel.

As mentioned earlier, free Mg²⁺ concentration of 10 mM abrogated depolarization-induced Ca²⁺ release[8,9]. We find two sites in RyR1 that get occupied at high Mg²⁺ but not low Mg²⁺: the high-affinity Ca²⁺ binding site and the EF-hand domain. Fractional occupancy must start somewhere between the two Mg²⁺ concentration studied, with the full occupancy observed here at 10 mM free Mg²⁺ probably corresponding to a lower concentration at physiological ionic strength[16,41]. This suggests that Mg²⁺ may operate at these two sites under physiological conditions, creating an inhibited conformation of RyR1. In addition, coordination of the inner helices by Mg²⁺ may translate into the stability of the closed conformation of RyR1 and resistance against reversible activation by DHPR. Interestingly, in RyR2, the low-affinity inhibition by Mg²⁺ (or Ca²⁺) only occurs at >10-fold higher concentration than in RyR1[10], indicating that the particular type of inhibition found here is unlikely to be of physiological relevance in cardiac muscle.

In sum, we posit that Mg²⁺ can inhibit the RyR1 by binding at the following different sites of RyR1: (a) the high-affinity Ca²⁺ activation site, where given its larger hydrated radius would keep the cavity in a "dormant" but Ca²⁺-accessible conformation, (b) the EF-hand domain, changing its orientation and priming it to form salt bridges with the neighboring subunit, and (c) the pore domain, coordinating the four inner helices in the closed conformation (see summary in Fig. 10). In the EC coupling mechanism, the depolarization-activated DHPR

activates its physically coupled RyR1s. We suggest that this action would increase separation of the S6 helices, at which point $Mg^{2+}$ would lose the coordination to S6, allowing $Ca^{2+}$ (and $Mg^{2+}$) permeation. $Ca^{2+}$ would then bind to the high-affinity site further activating the channel and subsequently, the $Ca^{2+}$ released into the cytoplasm would start to inactivate the channel limiting $Ca^{2+}$ release. In the absence of membrane depolarization, DHPR has an inhibitory effect on RyR1[42–44]. Thus, after skeletal muscle excitation ends and DHPR ceases to be activated, the inner helices of RyR1 would return to the closed conformation under direct action of the DHPR and recover their ability to coordinate $Mg^{2+}$.

## Methods

### Ethical statement
The study was performed in strict accordance with the recommendations in the Guide for the Care and Use of Laboratory Animals of the National Institutes of Health. Animals were handled according to the approved institutional animal care and use committee (IACUC) protocol #AD10001029 of Virginia Commonwealth University (to M.S.).

### Reagents
All chemicals were purchased from Thermo Fisher Scientific or Sigma-Aldrich except where indicated.

### Purification of rabbit skeletal muscle RyR1
Sarcoplasmic reticulum (SR) membranes were purified from young, 6 lbs., mixed gender New Zealand White rabbit back muscles through differential centrifugation as previously described[29]. 100 mg frozen membranes were thawed and solubilized in 10 ml of buffer-A containing 20 mM MOPS pH 7.4, 1 M NaCl, 9.2% (w/v) CHAPS, 2.3% (w/v) Phosphatidylcholine (PC; Sigma), 2 mM DTT and protease inhibitor cocktail with constant stirring for 15 min at 4 °C. The solubilized membranes were centrifuged at $100,000 \times g$ for 60 min and the pellet was discarded. Supernatant was layered onto a 10–20% (w/v) discontinuous sucrose gradient. Gradient was prepared in buffer-A containing 0.5% CHAPS and 0.125% PC (buffer-B). The layered sucrose gradient tubes were ultracentrifuged at $141,000 \times g$ for ~20 h at 4 °C to allow RyR1 separation. Fractions containing >95% pure RyR1 were pooled and further purified with HiTrap Heparin HP Agarose column (GE Healthcare), after a five-fold dilution step. RyR1 was eluted with buffer-B containing 0.635 M KCl after washing with 20 column volumes of buffer-B with 200 mM KCl. Peak fractions were flash-frozen and stored at −80 °C until nanodisc reconstitution and cryo-EM. 1.5–2 mg of RyR1 was purified from 100 mg of SR vesicles. RyR1 purity was estimated with 12.5% SDS-PAGE and negative staining with 0.75% Uranyl formate. Total protein concentration in purified microsomes and RyR1 fractions was measured with Quick Start Bradford Protein Assay (Bio-Rad).

### [³H] Ryanodine binding
RyR1 activity was estimated by measuring the extent of bound [³H] ryanodine in SR membrane vesicles isolated from rabbit skeletal muscle when incubated with free $Ca^{2+}$ (100 nM to 2 mM range) alone or in the presence of 1 mM free $Mg^{2+}$, 2 mM ATP, or 5 mM ATP-$Mg^{2+}$. Effect of free $Mg^{2+}$, ATP-$Mg^{2+}$, and ATP on activated RyR1 was studied by incubation of SR membrane vesicles with 50 μM $CaCl_2$ in the presence of 1 μM-5 mM $MgCl_2$, ATP-$Mg^{2+}$, or ATP. Concentrations of total divalent ion to be added to the reaction mixture were estimated in Maxchelator[45]. Preincubated membrane vesicles (30–50 μg) were allowed to bind 5 nM [³H] ryanodine (PerkinElmer) in a buffer containing 50 mM MOPS (pH 7.4), 0.15 M KCl, 0.3 mM EGTA, protease inhibitors, and 2 mM DTT for 1 h at 37 °C. Aliquots of samples were diluted seven-fold with an ice-cold wash buffer (0.15 M KCl) before placing onto Whatman GF/B filter papers in a vacuum-operated filtration apparatus. The radioactivity remaining in filter papers after washing three times with the wash buffer was estimated by liquid scintillation counting. Non-specific ryanodine binding was estimated in the presence of 25 μM unlabeled ryanodine (Tocris) and subtracted from the total binding. Data represent the mean specific [³H] ryanodine binding for three independent experiments.

### Reconstitution of RyR1 into nanodiscs
Reconstitution into nanodiscs was carried out as indicated previously[25]. Briefly, the plasmid pMSP1E3D1 (Addgene) encoding for MSP1E3D1 was expressed in *E.coli* and purified using the manufacturer's instructions. RyR1-nanodiscs were obtained by mixing purified RyR1, MSP1E3D1 and POPC (Avanti polar lipids) at a 1:2:50 molar ratio. The mixture was incubated for 1 h 30 min at 4 °C before overnight dialysis in a CHAPS-free buffer (20 mM MOPS pH 7.4, 0.635 M KCl, 2 mM DTT) containing 5 and 11.6 mM $MgCl_2$ in case of ACP/L$Mg^{2+}$ and ACP/H$Mg^{2+}$ respectively. In the ACP/EGTA condition, 1 mM EGTA and 1 mM EDTA were included. Maxchelator[45] was used to compute free $Mg^{2+}$. The channels' integrity was confirmed using negative staining prior to cryo-EM.

### Cryo-EM grid preparation and data acquisition
Cryo-EM grids were cleaned with a customized protocol[46] prior to glow-discharge. The dialyzed RyR1-nanodisc preparations were incubated for 30 min with either 5 mM ACP/5 mM $Mg^{2+}$ (~1 mM free $Mg^{2+}$; ACP/L$Mg^{2+}$ dataset) or with 1 mM ACP/ 11.6 mM $Mg^{2+}$ (~10 mM free $Mg^{2+}$; ACP/H$Mg^{2+}$ dataset) before being plunge frozen. Aliquots of 1.25–1.5 μl RyR1-nanodisc were applied onto each side of glow-discharged C-Flat − 1.2/1.3 Au holey-carbon (Protochips, NC) or Ultra-Aufoil −1.2/1.3 holey-gold (Quantifoil, Germany) with 300 TEM mesh. The grids were blotted with Whatman 540 filter paper for 1–1.5 s in a Vitrobot™ Mark IV (Thermo Fisher Scientific) and rapidly plunged into liquid ethane. The sample quality was initially assessed on a Tecnai F20 (Thermo Fisher Scientific) electron microscope. Cryo-EM data was collected in multiple sessions on a Titan Krios transmission electron microscope (Thermo Fisher Scientific) operated at 300 kV under super-resolution (RyR1-ACP/L$Mg^{2+}$) or counting mode (RyR1-ACP/ H$Mg^{2+}$), with a K2 or K3-Summit detector (Gatan) as shown in Supplementary Table 1. A Gatan Quantum Energy Filter (GIF) with a slit width of 20 eV was employed in all three datasets. Datasets were collected in automated mode with the Latitude program (Gatan) with a total electron dosage of 60–70 e⁻/Å² applied across 50–60 frames. Supplementary Table 1 summarizes the image acquisition parameters.

### Single-particle image processing
Gain-reference normalization, frame alignment, dose-weighting, and motion correction of the collected movie stacks was carried out with Motioncor2[47] or patch motion correction employed in cryosparc3.2. Contrast transfer function parameters were estimated from non-dose weighted motion-corrected summed images using CTFFIND 4.0[48] or Gctf[49]. All subsequent image processing operations were carried out using dose weighted, motion-corrected micrographs in RELION 3.0 or cryosparc3.2[50]. 2D class average templates for auto-picking were generated by reference-free 2D classification of 1000 manually picked particles. Autopicked particles with ethane and hexagonal ice-contaminated areas and junk particles were removed prior to 2D classification. Particle image stacks required for the reconstructions focused on domains spanning from CD to TMD (residues 3668–5037) and quarter (one-fourth) sub-volumes of RyR1 were generated using a signal subtraction procedure (employed in relion_project) which was combined with symmetry expansion[51] for the one-fourth sub-volume with relion_particle_symmetry_expand tool. The reported resolutions of the cryo-EM maps are based on FSC = 0.143 criterion[52]. Local resolution was estimated with ResMap[53]. Pixel size calibration of post-processed EM maps of the datasets were carried out using real space correlation metric of UCSF Chimera[54] using a published RyR1 cryo-EM

map[17]. Pixel size of 1.36 and 0.86 Å, were obtained for RyR1-ACP/LMg$^{2+}$ and RyR1-ACP/HMg$^{2+}$, respectively. Two datasets collected under identical imaging conditions for the RyR1-ACP/LMg$^{2+}$ were merged after a real space calibration test. Image processing schemes are summarized in Supplementary Figs. 1 and 2.

## Identification of the Mg$^{2+}$ density in the ATP binding cavity

Cryo-EM map and model of RyR1-ACP/Mg$^{2+}$ were aligned and resampled to the RyR1-ACP/EGTA map with fit in map and vop resample tools in UCSF Chimera, respectively. Localized maps around the ATP molecule were calculated by extracting 3D-density boxes with a selection radius 10 Å from the ACP using the Phenix Map Box tool[55]. Difference maps in real space for the localized regions of RyR1 were calculated with the vop subtract tool in UCSF Chimera.

## Model building and structure refinement

The cryo-EM based atomic model of RyR1-ACP/EGTA (PDB ID: 7K0T) was taken as the initial model for model building. Using the Chimera Fit in map tool, the best resolved EM map from the relevant dataset was docked with a RyR1 tetramer. An iterative approach of amino acid fitting was used to enhance the local density fit of a protomer in Coot 0.9.5.8[56], which was alternated with real space refinement in PHENIX 1.2rc4-4425[57]. Real space refinement was carried out with secondary structure and Ramachandran restraints. The consensus reconstructions were used to assemble the tetramer models before real space refinement with NCS restraints. Comprehensive model validation of the models was carried out with Phenix.

## Pore radius calculation

Pore radius measurements were carried out with the refined atomic model coordinates of the RyR1 pore region (residues 4821–5037) with the HOLE program[58]. Dot surfaces representing the channel ion permeation pathway were generated with HOLE (implemented in Coot 0.9.5.8) and reformatted (to .bld files) that enabled visualization in UCSF chimera.

## Molecular dynamics simulations

To further investigate the interactions between various cations and the protein in alignment with our experimental findings, we conducted all-atom molecular dynamics (MD) simulations in explicit solvent. Structural models for RyR1/LMg$^{2+}$, RyR1/HMg$^{2+}$ and RyR1/Ca$^{2+}$ were constructed based on cryo-EM structures (PDB IDS: 7K0S, 7UMZ, 7TDH, respectively) encompassing residues in the pore domain including S5, pore helix, selectivity filter, and S6 (residues 4835–4956). The protonation states of ionizable residues (Lys, Arg, Asp, and Glu) at neutral pH were assigned according to the pK$_a$ prediction using the PROPKA software[59]. Each model was embedded in a pre-equilibrated 1-Palmitoyl-2-Oleoyl-sn-Glycero-3-Phosphocholine (POPC) lipid environment to mimic experimental conditions and solvated with TIP3P water[60]. Counterions were added to neutralize the overall charge of the systems, and a salt concentration of 0.6 M KCl was maintained using VMD's Autoionize plugin[61].

Model systems 1–6 were designed to assess the stability of the putative cation identified in the D4945 site under different structural states. In models 1, 2, and 3, hexahydrated Mg$^{2+}$ was placed at the position dictated by the coordinates of the point of highest density within the pore axis in the cryo-EM structure of RyR1-ACP/HMg$^{2+}$. To compare to another common divalent cation, Mg$^{2+}$ was replaced with heptahydrated Ca$^{2+}$ in models 4, 5, and 6. We employed enhanced force field parameters and topology of a hydrated form of Mg$^{2+}$ and Ca$^{2+}$ as developed by Yoo and Aksimentiev[62]. A summary of the simulation systems is shown in Supplementary Table 2.

All-atom MD simulations were carried out with periodic boundary conditions. A distance cutoff of 12 Å was used for calculating nonbonded interactions. Electrostatic interactions were calculated with

particle mesh Ewald summation via fast Fourier transform, including van der Waals interactions, with a switching distance of 10 Å. Langevin dynamics at a constant temperature of 298 K was used with a damping coefficient of 1 ps$^{-1}$. The pressure was kept constant at 1 atm using the Nosé−Hoover Langevin piston method, with a piston period of 200 fs and a damping time of 50 fs. Energy minimization was performed to remove bad contacts between atoms. Restrained MD simulations were employed to relax the structural strains of the model systems. In the first stage, the protein and the cation were initially kept fixed to their initial positions while water, lipid, and counterion were allowed to equilibrate for about 1 ns. Then, the systems were equilibrated by 1 ns MD simulations with position restraints of protein atoms and the cation. During this step, a harmonic potential of force constant decreases from 1 to 0.1 kcal/(mol Å$^2$). Subsequently, MD simulations were performed for each system under NPT ensemble. To compensate for possible edge effects of the truncated regions in the model, a softness in the restraints was applied to Cα atoms of the protein of the last four residues of the N- and C-termini. The CHARMM36 force field parameter sets[63] were applied for protein, lipid, and counterions. The simulated systems were constructed using TCL language in VMD 1.9.3[61]. All simulations were carried out for 100 ns using the NAMD 2.12 software[64]. Three independent runs of MD simulations were performed for each model system.

Analyses of MD trajectories, including hydrogen bond and ion displacement were carried out using TCL scripts in VMD. Unless otherwise specified, analysis of MD trajectories was carried out during the last 20 ns simulation. This was primarily to make sure that the systems had reached thermodynamic equilibrium and that structural features extracted from the trajectory were adequately characterized with the least deviation.

## Binding energy calculations from density function theory

To assess the relative stability of the metal (M) in the D4945 site, we explored the binding energy of the [M(H$_2$O)$_n$(D4945)$_4$]$^{2-}$ complex using a density functional theory (DFT) method. The [M(H$_2$O)$_n$(D4945)$_4$]$^{2-}$ models containing the four D4945 residues and the hexahydrated Mg$^{2+}$, [Mg(H$_2$O)$_6$]$^{2+}$, or heptahydrated Ca$^{2+}$, [Ca(H$_2$O)$_7$]$^{2+}$, were taken from MD snapshots of the RyR1/LMg$^{2+}$+Mg$^{2+}$ and RyR1/HMg$^{2+}$+Mg$^{2+}$ systems (models 1, 2, 4 and 5). We performed a single-point calculation on these models using the DFT method at the B3LYP level and a 6−31 + G(d,p) basis set[65] to determine the binding energy of the complex. The solvent effect was implicitly included using the CPCM model. The binding energies (ΔE$_{bind}$) were obtained according to the equation:

$$\Delta E_{bind} = E_{AB} - (E_A + E_B) \tag{1}$$

where E$_{AB}$ is the energy of the [Mg(H$_2$O)$_6$(D4945)$_4$]$^{2-}$ or [Ca(H$_2$O)$_7$(D4945)$_4$]$^{2-}$ complexes, and E$_A$ and E$_B$ are the energies of subsystems [Mg(H$_2$O)$_6$]$^{2+}$ or [Ca(H$_2$O)$_7$]$^{2+}$ and (D4945)$_4$$^{4-}$, respectively. Total energies of the complex (E$_{AB}$) and subsystems (E$_A$ and E$_B$) were computed separately. Moreover, we further examined an impact of the optimized structure on the binding energy of the complex. The geometry optimization on these models was performed using the PM3 method[66]. Subsequently, binding energy calculations on the optimized structure were carried out with the same level of theory. Convergence was achieved when the maximum force acting on any atom fell below 0.001 Hartree/Bohr. The maximum allowed displacement was set at 0.005 Å. All DFT computations were performed using the Gaussian 09 software package[67].

## Statistical analysis

Data are presented as means±SEM, with the number of independent experiments indicated in the figure legends. SEMs calculated for each data point for mean specific [$^3$H] Ryanodine binding are reported. [$^3$H]

Ryanodine binding data were fitted using nonlinear regression analysis in GraphPad Prism 5.0. Figures were generated using PyMOL 2.5.4, ChimeraX 1.3, and CorelDrawX7. Movies were generated using PyMOL, ChimeraX, VMD 1.9.3, Windows Movie Maker 2012, and ffmpeg in Linux. Graphs were generated with Graphpad Prism and OriginPro 2015.

## Reporting summary

Further information on research design is available in the Nature Portfolio Reporting Summary linked to this article.

## Data availability

The data that support this study are available from the corresponding authors upon request. The cryo-EM maps generated in this study have been deposited in the Electron Microscopy Data Bank under accession codes EMD-22615 (RyR1-ACP/LMg$^{2+}$); and EMD-26610 (RyR1-ACP/HMg$^{2+}$). The atomic coordinates generated in this study have been deposited in the Protein Data Bank under accession codes: 7K0S (RyR1-ACP/LMg$^{2+}$); and 7UMZ (RyR1-ACP/HMg$^{2+}$). Previously published protein structure data used for analysis in this study are available in the Protein Data Bank under accession codes: 5TAL (RyR1-ATP/Ca$^{2+}$/Caffeine); 7TDH (RyR1-ACP/Ca$^{2+}$ open); 5TB0 (RyR1-EGTA); 7K0T (RyR1-ACP/EGTA); and 7TDG (RyR1-ACP/Ca$^{2+}$ inactivated). The source data underlying Figs. 1 and 4 is provided as a Source Data file.

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

## Acknowledgements

Cryo-grid preparation and screening were carried out at the Cryo-EM Unit at Virginia Commonwealth University (VCU) supported by the VCU School of Medicine and M.S.'s funds. Cryo-EM data collection was carried out at the Frederick National Laboratory for Cancer Research supported by contract HSSN261200800001E where we thank Drs. Thomas Edwards, Ulrich Baxa, and Adam Wier for cryo-EM data collection, and at the Molecular Electron Microscopy Core Facility at the University of Virginia (UVA). Supported by NIH R01 AR068431 and U24GM116790 (to M.S.), and Thailand Science Research and Innovation Fund Chulalongkorn University HEA662300080 (to P.S.).

## Author contributions

A.R.N., A.H.W., P.C.H., and Y.H. performed protein purification and cryo-grid preparation; K.D. contributed with data collection; A.R.N., J.J.L., and A.H.W. performed image processing; A.R.N. performed model building and radioligand binding assays; P.S. and W.R. carried out molecular dynamics, quantum chemical calculations, and validation; M.S. conceived, designed, and supervised all cryo-EM experiments; G.D.L. provided valuable intellectual input; M.S., P.S., A.R.N., and G.D.L. interpreted the data and wrote the manuscript; A.H.W. edited the manuscript.

## Competing interests

The authors declare no competing interests.
