## [Peer Review File · Nature Communications]

Interplay between Mg^{2+} and Ca^{2+} at multiple sites of the ryanodine receptorReviewer #1 (Remarks to the Author):

In my opinion the manuscript "Interplay between Mg²⁺ and Ca²⁺ at multiple sites of the ryanodine receptor" reports noteworthy results that significantly advance the field. The combination of functional, structural and MD methods provides convincing insights into the mechanism of Mg²⁺ inhibition of RyR1 channels. Being an expert on MD simulations of ion channels, the evaluation of the soundness of the methodology is on this part only.

Force field parameters for divalent ions, such as studied here can be challenging. The authors are aware of this challenge and employed enhanced force field parameters for their simulation.

Reassuringly, the DFT simulations carried out provide additional support for the stable binding (inhibition) of Mg²⁺ at D4945, observed in MD simulations, thus supporting the soundness of the used FF parameters.

The methods section provides sufficient detail to reproduce the modelling performed. Thus, I have no criticism or suggestions for the computational part of the manuscript.

Reviewer #2 (Remarks to the Author):

In the manuscript entitled "Interplay between Mg²⁺ and Ca²⁺ at multiple sites of the ryanodine receptor", Nayak et al. revealed the inhibitory mechanism of RyR1 by Mg²⁺, a novel and interesting finding in RyR field. With the revisions I suggest below, I recommend the manuscript be published in this journal.

1. Line 126-129, For clarity, it would be good to show the comparison of the pore distances by a supplementary figure.
2. Line 143-147, The hydrogen bond formed by R4944-E4942' was not indicated in the Figure 3B, and it seems like that the E4942' is far from the R4944 in the ACP/EGTA structure. For clarity, I recommend to indicate the residues in all the panels of the figures throughout the manuscript. In addition, it would be better to show the "D4945-R4944-D4938" as "D4945'-R4944-D4938", indicating that they are located at different protomer.
3. It is interesting to see the conformational changes in the EF hand upon high Mg²⁺ condition, the interactions between EF hand and S2-S3 loop may serve as another transducer for transferring the allosteric information from the cytoplasmic domain of RyR1 to the TMD. Whether this conformational change in this region during channel gating in RyR2 is similar with that in RyR1? Whether the inhibitory mechanism of RyR2 by Mg²⁺ is the same as that of RyR1? A discussion for these issues may be better for understanding the functional mechanisms of RyRs.
4. The letter in the figure legend should be in lower case.

Dear Review panel,

We appreciate the constructive comments to our manuscript. Please find below our point-by-point response, with the corresponding changes in the manuscript highlighted in yellow. We also updated the format according to the Journal's guidelines.

Reviewer #1

In my opinion the manuscript "Interplay between Mg²⁺ and Ca²⁺ at multiple sites of the ryanodine receptor" reports noteworthy results that significantly advance the field. The combination of functional, structural and MD methods provides convincing insights into the mechanism of Mg²⁺ inhibition of RyR1 channels. Being an expert on MD simulations of ion channels, the evaluation of the soundness of the methodology is on this part only.

Force field parameters for divalent ions, such as studied here can be challenging. The authors are aware of this challenge and employed enhanced force field parameters for their simulation. Reassuringly, the DFT simulations carried out provide additional support for the stable binding (inhibition) of Mg²⁺ at D4945, observed in MD simulations, thus supporting the soundness of the used FF parameters.

The methods section provides sufficient detail to reproduce the modelling performed. Thus, I have no criticism or suggestions for the computational part of the manuscript.

We thank reviewer #1 for revising the manuscript and examining in depth the MD section. We are pleased that there were no criticisms.

Reviewer #2

In the manuscript entitled "Interplay between Mg²⁺ and Ca²⁺ at multiple sites of the ryanodine receptor", Nayak et al. revealed the inhibitory mechanism of RyR1 by Mg²⁺, a novel and interesting finding in RyR field. With the revisions I suggest below, I recommend the manuscript be published in this journal.

We thank reviewer #2 for reviewing the manuscript and for the constructive and helpful comments. Please find below our responses.

1. Line 126-129, For clarity, it would be good to show the comparison of the pore distances by a supplementary figure.

We appreciate the suggestion and in fact we expanded the measurement of distances to all structures presented in Fig. 3c. These measurements give a clearer picture of the similarity of the closed channel at the level of I4937 under any closed-state condition, while only presence of Mg²⁺ induces tightening of the cytoplasmic portion of S6. These distances are now presented as part of Figure 3c. We have moved the content of Lines 126-129 further

down at the mention of Figure 3c (lines 147-151).

2. Line 143-147, The hydrogen bond formed by R4944-E4942' was not indicated in the Figure 3B, and it seems like that the E4942' is far from the R4944 in the ACP/EGTA structure. For clarity, I recommend to indicate the residues in all the panels of the figures throughout the manuscript. In addition, it would be better to show the "D4945-R4944-D4938" as "D4945'-R4944-D4938", indicating that they are located at different protomer.

We appreciate the observation. The panel for ACP/EGTA has now been updated with the final structure deposited in the pdb database. Also, thanks for the suggestions. We now distinguish the residues from the different chains and indicate the residues in all the figure's panels throughout the manuscript.

3. It is interesting to see the conformational changes in the EF hand upon high Mg^{2+} condition, the interactions between EF hand and S2-S3 loop may serve as another transducer for transferring the allosteric information from the cytoplasmic domain of RyR1 to the TMD. Whether this conformational change in this region during channel gating in RyR2 is similar with that in RyR1? Whether the inhibitory mechanism of RyR2 by Mg^{2+} is the same as that of RyR1? A discussion for these issues may be better for understanding the functional mechanisms of RyRs.

We agree with the reviewer that a thorough comparison with the RyR2 under similar conditions will be important to understand better the correlations between structure and function for the two isoforms. However, at present there is no structural data of RyR2 resolved under conditions of high Mg^{2+} , and carrying out such determination would go beyond the scope of the current study. So far, structures of RyR2 have been resolved either in the presence of EGTA, or in the presence of activating Ca^{2+} concentrations, usually around 30 μM . This is far from the low mM inhibiting concentrations of divalent cation used in our structural determinations of RyR1. Importantly, the low affinity inhibition of RyR2 by Mg^{2+} or inactivation by Ca^{2+} only takes place at >10-fold higher concentration than in RyR1 (see ref 10, Laver et al, 1997), so it is less relevant in the physiological context. We now explicitly discuss this at the end of the second last paragraph in the Discussion (lines 396-398). We also added "RyR1" to the section heading: "Conformational change of the EF-hand domain **of RyR1** at high Mg^{2+} " in line 208 for further clarity.

4. The letter in the figure legend should be in lower case.

Thanks for pointing this out. We have changed the case of the panels and reflect so in the figure legend.

Reviewer #2 (Remarks to the Author):

I am satisfied by the corrections to the manuscript and recommend publication.